# BEHAVIOUR-AWARE MULTIMODAL VIDEO SUMMARIZATION: CROSS-MODAL INTEGRATION FOR HUMAN-CENTRIC CONTENT ANALYSIS

## ABSTRACT

Video summarization remains a challenging task in capturing the complex interplay of visual dynamics, spoken content, and behavioural cues that collectively shape viewer understanding in human-centric videos. Human communication is inherently multimodal; however, existing approaches in video summarization either rely solely on visual features or rudimentary text-visual combinations, neglecting critical audio prosodic patterns and their interactions. Crucially, the synchronous behavioural signals that convey emotional expression and communicative intent are not considered entirely. In this paper, we present a behaviour-aware multimodal framework for video summarization that explicitly models synchronized behavioural cues across visual, audio, and textual modalities through a transformer-based architecture with cross-modal attention mechanisms. Our approach integrates CLIP visual embeddings enhanced with facial movement detection and emotional transitions, HuBERT audio features enriched with prosodic patterns including pitch variations and voice quality measures, and RoBERTa textual embeddings that preserve narrative flow and discourse structure. We employ heuristic-based behavioural cue detection methods combined with large language model-guided extractive summarization to generate pseudo-ground truth references that capture both semantic importance and behavioural salience. Extensive evaluations on the ChaLearn First Impressions dataset demonstrate substantial improvements over state-of-the-art methods, achieving a 33.2% increase in F1-score over CLIP-It and 7.3% over recent multimodal approaches. Comprehensive ablation studies confirm the effectiveness of behavioural cue integration, with each modality contributing complementary insights for capturing communicatively significant moments in interview-style videos.

## 1 INTRODUCTION

The rapid proliferation of video content across diverse platforms such as education, social media, professional interviews, and journalism has heightened the demand for automated video summarization techniques capable of distilling complex, multimodal videos into concise and meaningful summaries. Traditional summarization methods often rely on individual cues, such as scene transitions or frame salience (Otani et al., 2019; Zhang et al., 2016), which fail to capture the rich interplay of visual, auditory, and textual modalities inherent in modern videos. This limitation is particularly pronounced in videos rich in human interaction, such as interviews, where coordinated visual gestures, spoken narratives, and ambient audio convey behavioural and contextual information (Evangelopoulos et al., 2013). Multimodal video summarization seeks to address this gap by integrating multiple modalities to produce semantically rich and contextually relevant summaries. However, existing approaches face significant challenges, including effectively modeling cross-modal interactions, addressing modality misalignment, and overcoming the scarcity of annotated datasets (Argaw et al., 2024; Qiu et al., 2023). While recent advancements, such as VSL (Lynch et al., 2024) and CFSum (Guo et al., 2025), have made progress in integrating visual, audio, and textual modalities, they still fall short in capturing the behavioural features that are crucial for human-centric videos.

In this paper, we present a novel multimodal framework for summarizing interview videos, emphasizing behavioural cues (gestures, vocal prosody) alongside audio and textual data from transcripts.

Unlike prior methods that process modalities independently (Narasimhan et al., 2022; Evangelopoulos et al., 2013), our approach gets inspiration from transformer-based architecture with cross-modal attention (Vaswani et al., 2017) to integrate visual, auditory, and textual features, highlighting communicative significance and capturing semantic and emotional contexts. An autoregressive decoding strategy ensures temporally coherent segments, mitigating class imbalance in binary classification (Narasimhan et al., 2021) and reducing redundancy in frame-based scoring (Narasimhan et al., 2022; He et al., 2023; Zhang et al., 2016). The framework features a preprocessing pipeline with a forced alignment technique (McAuliffe et al., 2017; Argaw et al., 2024) for millisecond-precision synchronization, modality-specific encoders, and a cross-modal attention mechanism to prioritize relevant features. Due to the lack of human-annotated summaries, we propose a two-stage method: heuristic-based detection of behavioural cues (facial expressions, prosodic patterns, gestural emphasis) followed by integration with timestamped transcripts as metadata to guide LLMs in generating pseudo-ground truth summaries (Argaw et al., 2024; Moinul Islam et al., 2025), combining semantic content and behavioural significance. Speech-to-text models (Radford et al., 2023; Baevski et al., 2020) provide timestamped transcripts, enabling LLMs to select key sentences with preserved temporal markers for video segment mapping.

Standard video summarization datasets, such as SumMe (Gygli et al., 2014) and TVSum (Song et al., 2015), focus on action-oriented content (e.g., sports, news, documentaries) with limited human interaction, which contrasts with our behaviour-aware approach. We evaluate our framework using the ChaLearn First Impressions dataset (Ponce-López et al., 2016), comprising high-quality interview-style videos with single speakers in controlled settings. This dataset offers: (1) consistent single-speaker format for precise behavioural analysis; (2) rich multimodal cues (facial expressions, gestures, and vocal variations); (3) clear audio-visual synchronization; (4) transcript availability; and (5) diverse emotional and communication styles. Unlike action-centric datasets where visual salience prevails, ChaLearn's emphasis on subtle behavioural signals aligns with our framework's design, making it ideal for evaluation. Our work makes the following key contributions:

1. We introduce a novel transformer-based multimodal summarization framework with cross-modal attention that explicitly models synchronized behavioural cues, such as gestures and vocal prosody, across visual, audio, and textual modalities. Unlike recent state-of-the-art methods (Lynch et al., 2024; Guo et al., 2025), which focus on general content relevance, our approach emphasizes communicative intent by prioritizing behaviour-aware features, which is crucial for interview video summarization.

2. We advance multimodal feature representation by extracting behaviour-specific features: (a) CLIP visual embeddings enhanced with facial movements and emotional transitions, (b) HuBERT audio embeddings capturing prosodic patterns, and (c) contextual text representations preserving narrative flow. This refined approach contrasts with existing methods (Apostolidis et al., 2021; Argaw et al., 2024) that rely on generic multimodal fusion and fail to capture behavioural cues.

3. We contribute a comprehensive evaluation strategy that integrates text and video-based metrics, validated on the ChaLearn First Impressions dataset. Adopting the pseudo-ground truth generation techniques demonstrated in Argaw et al. (2024); Moinul Islam et al. (2025), our approach enables robust comparisons across summarization methods through LLM-generated reference summaries, with our framework outperforming state-of-the-art models such as CLIP-It (Narasimhan et al., 2021) by 33.2%, and Argaw et al. (2024) by 7.3% in F1-score. This demonstrates the effectiveness of behaviour-aware summarization in producing high-quality, contextually rich summaries.

4. By addressing the limitations of existing multimodal approaches and introducing a behaviour-aware perspective, our framework sets a new standard for video summarization, with potential applications in human-computer interaction and affective computing.

## 2 RELATED WORKS

Video summarization has evolved from unimodal approaches, which rely solely on visual features, to multimodal frameworks that integrate visual, auditory, and textual modalities to capture richer semantic and contextual information.

Unimodal video summarization focuses on visual features, such as keyframes, scene transitions, or object dynamics, using heuristic, statistical, or deep learning-based methods for frame importance scoring or sequence modeling. These approaches typically ignore complementary modalities such as audio and text, limiting contextual and emotional richness. Otani et al. (2017) proposed a clustering-based method that utilizes deep semantic features extracted from video segments to produce coherent and accessible summaries. Apostolidis et al. (2020) introduced an unsupervised GAN-based model augmented with an actor-critic framework, improving content representation without the need for labeled data. Similarly, Zhou et al. (2018) employed deep reinforcement learning to frame the summarization task as a sequential decision-making process, optimizing frame selection via diversity and representativeness rewards. Feng et al. (2018) proposed a memory-augmented network for preserving temporal structure while enabling sparse frame extraction. Yuan & Zhang (2022) extended reinforcement-based strategies by refining shot-level semantics, improving coherence in summary generation. Zhang et al. (2019b) emphasized temporal dependencies using a dilated temporal relational adversarial network, while Messaoud et al. (2021) introduced query-aware summarization via hierarchical pointer networks to align outputs with user intent. Leveraging attention mechanisms, VASNet scores frames based on temporal dependencies, achieving coherent summaries (Fajtl et al., 2019). Mahasseni et al. (2017) proposed an adversarial LSTM-based framework that balances generative and discriminative objectives to produce visually diverse and representative summaries. Building on this, Yuan et al. (2019) introduced Cycle-SUM, which enforces cycle consistency through adversarial training with LSTMs to improve temporal coherence.

Multimodal video summarization integrates visual, auditory, and textual modalities to produce semantically rich and contextually relevant summaries, addressing the limitations of unimodal approaches by capturing narrative structure, emotional undertones, and contextual importance. Recent advancements leverage attention mechanisms, memory-augmented networks, and large language models (LLMs) to enhance summary coherence, personalization, and cross-modal alignment. A robust body of work demonstrates the effectiveness of combining these modalities, though explicit focus on human behaviour-aware summarization remains limited. Early frameworks, such as MM-VS (Evangelopoulos et al., 2013), combined visual and audio cues for movie summarization using saliency-based fusion. Raventos et al. (2015) proposed a framework for soccer videos using audio and visual descriptors, though it omitted textual data. More recent approaches have incorporated all three modalities. For instance, Lynch et al. (2024) demonstrated how vision-language models align multimodal features for accurate summarization, while V2XUM-LLM (Hua et al., 2025) uses temporal prompt tuning with LLMs to enhance video-text alignment. The VSL framework (Lynch et al., 2024) personalizes summaries using video, audio, and closed captioning. Similarly, CFSum (Guo et al., 2025) employed a coarse-fine fusion approach, emphasizing audio's role alongside visual and textual features. Apostolidis et al. (2021) combined local and global attention with positional encoding to model temporal dependencies, ensuring contextually coherent summaries. Argaw et al. (2024) proposed a transformer-based framework that integrates visual and textual features via cross-modal attention, with a masking strategy for text-less scenarios. Psallidas et al. (2021) focused on user-generated videos, using audio and visual features to create dynamic summaries, highlighting the underutilized potential of auditory cues. Zhao et al. (2022) introduced a hierarchical multimodal transformer by integrating visual and audio modalities. Palaskar et al. (2019) developed a language-driven framework for abstractive summarization of instructional videos, enabling user-specific summaries. Lynch et al. (2024) further advanced personalization by incorporating user preferences, such as genres, into multimodal summarization. Zhu et al. (2023) proposed a topic-aware summarization task, generating multiple summaries based on different topics using a multimodal transformer. Zhao et al. (2022) introduced dynamic sampling to capture inter-frame variations, enhancing multimodal integration. Targeting instructional videos (TL;DW), Narasimhan et al. (2022) integrated visual content with textual metadata via cross-modal saliency to prioritize task-relevant moments . MultiSum (Qiu et al., 2023) provides a dataset and methods for multimodal summarization, combining visual frames with textual transcripts. CLIP-It (Narasimhan et al., 2021) and VideoBERT (Sun et al., 2019) utilize vision-language pretraining with cross-modal attention to align visual and textual modalities, improving summarization quality.

Despite these advances, most multimodal approaches prioritize content relevance over explicit modeling of human behavioural cues, such as gestures, facial expressions, or vocal intonations. For example, while Psallidas et al. (2021) uses audio features that may capture speech tone, it does not explicitly target behavioural nuances. Similarly, Lynch et al. (2024) focuses on user preferences rather than behavioural signals within the video content. Some works indirectly address behaviour-

related aspects. For instance, Ma et al. (2023) explored human-machine collaboration using pupillary response signals to guide attention-based summarization, reflecting viewer engagement. However, this approach is unimodal and does not integrate audio or textual cues.

Our proposed framework addresses this gap by explicitly integrating visual dynamics, audio prosody, and textual transcripts to generate behaviour-aware video summaries. Unlike existing methods that focus on general content or user preferences, our approach employs cross-modal attention mechanisms to model behavioural cues, such as vocal intonations and gestures, ensuring summaries reflect both semantic content and emotional nuance. By leveraging LLM-based supervision, the framework enhances contextual understanding, particularly for human-centric videos. This distinguishes our work from prior approaches, such as VSL (Lynch et al., 2024), which prioritizes personalization, or MFST (Park et al., 2022), which focuses on multimodal frame-scoring without explicit behavioural modeling.

In summary, the field of multimodal video summarization has seen significant progress, with frameworks such as VSL, CFSum, and others demonstrating the power of integrating visual, audio, and textual modalities. However, the explicit incorporation of human behavioural cues remains a largely unexplored frontier. Our proposed framework advances the state-of-the-art by focusing on behaviour-aware summarization that links video, audio and text modalities, while offering a novel approach to capturing the emotional and contextual richness of human-centric videos.

## 3 METHODOLOGY

The proposed framework employs a transformer-based encoder-decoder architecture to process and combine multimodal features for summarization and utilizes LLMs to generate behaviour-aware pseudo-ground truth summary videos for the evaluation purpose, as shown in Figure 1.

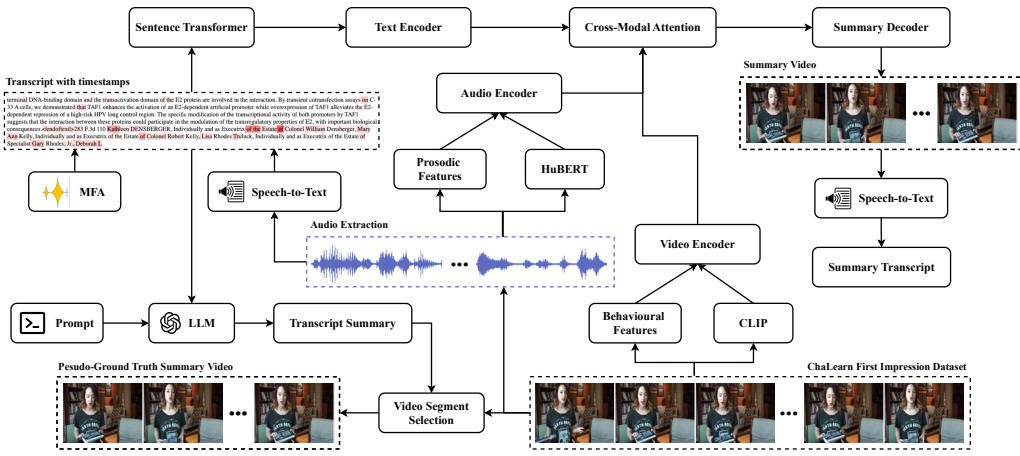

Figure 1: **Architecture of our proposed behaviour-aware multimodal video summarization framework.** The diagram illustrates the dual-pipeline approach: (1) pseudo-ground truth generation (bottom) using LLM-guided extractive summarization to create reference summaries from timestamped transcripts, and (2) the multimodal summarization framework (top) integrating three parallel processing streams through modality-specific encoders. The framework employs cross-modal attention to fuse representations from all modalities and uses an autoregressive summary decoder to generate temporally coherent video summaries.

**Data Sources.** This study utilizes the ChaLearn First Impressions dataset (Ponce-López et al., 2016), a publicly available collection of 10,000 high-quality video clips featuring 7,138 unique subjects speaking English during job interview tasks, averaging 15 seconds (range: 8–20 seconds) at approximately 24 frames per second (fps), with mono audio at 16 kHz and transcripts averaging 38 words (around 152 characters) per clip. Originally designed for personality trait recognition, the dataset captures diverse behavioural cues, making it ideal for behaviour-aware multimodal video summarization. A stratified subset of 1,500 clips is selected for the evaluation to ensure diversity in

age, gender, and behavioural expressions. The detailed information on the preprocessing step can be found in Section A.1 of the Appendix.

## 3.1 Multimodal summarization framework

Let $V = \{F_1, F_2, \ldots, F_n\}$ represent a video as a sequence of $n$ frames $F_i$ sampled every $\Delta$ seconds. Let $A = \{W_1, W_2, \ldots, W_p\}$ denote the audio waveform segmented into $p$ frames $W_i$, while $T = \{T_1, T_2, \ldots, T_k\}$ corresponds to the transcribed text of $V$ as a sequence of $k$ sentences $T_i$ (i=1 to k). Given the input $\{V, A, T\}$, the aim is to generate a summary video $Y = \{Y_1, Y_2, \ldots, Y_m\}$, where $Y_i$ are selected frames of $V$. This process involves three main components: multimodal processing (visual, audio and text related modalities), cross-modal integration (through cross-modal attention) and, finally, summary generation. These components are detailed below.

**Visual processing.** The visual processing pipeline encodes behavioural and semantic moments for concise video summaries, integrating traditional and vision language-based feature extraction with transformer-based video encoding to capture expressive dynamics. Traditional approaches identify frames with significant behavioural signals using MediaPipe Pose (Lugaresi et al., 2019) to track facial landmarks (e.g., nose, eyes) as 3D coordinates. Facial movement, such as head nods signaling engagement, is quantified by the mean Euclidean distance between landmarks across consecutive frames. In the same spirit as (Otani et al., 2017), an adaptive threshold, the mean plus standard deviation over a 10-frame window, is used to flag expressive frames. Next, emotional shifts, such as neutral to happy transitions, are detected using DeepFace (Serengil & Ozpinar, 2024), which classifies emotions (e.g., happy, sad, neutral) over the same window, marking frames with distinct changes.

We employ Contrastive Language-Image Pre-training (CLIP) (Radford et al., 2021) to obtain a visual embedding for each frame. CLIP's vision transformer (ViT) produces 512-dimensional embeddings for frames sampled at 1 fps, resized to $224 \times 224$ pixels (RGB, $[0, 1]$ scale). Head movement scores and emotion labels are concatenated with CLIP embeddings to form enhanced visual tokens $\{v_1, v_2, \ldots, v_n\}$, capturing dynamics such as nods during confident statements. The token sequence, augmented with start-of-sequence (SOS) and end-of-sequence (EOS) tokens and positional encodings (Vaswani et al., 2017), is processed by a video encoder. Multi-head self-attention enables temporal reasoning:

$$\{\hat{v}_i\}_{i=0}^{n+1} = \textbf{V-Encoder}\big(\{\texttt{SOS}, v_1, \ldots, v_n, \texttt{EOS}\}\big) \tag{1}$$

The video encoder transforms per-frame embeddings into temporally coherent representations combining static semantic content and behavioural dynamics.

**Audio processing.** Given an audio stream as a sequence of 16 kHz mono waveform segments, this pipeline extracts semantic embeddings and behavioural cues. YAAPT (Kasi, 2002) estimates fundamental frequency ($F_0$), averaging pitch contours over a 10-frame window to detect prosodic expressiveness (e.g., rising pitch for emphasis). Missing values are interpolated linearly. OpenS-MILE's eGeMAPS (Eyben et al., 2010) computes standardized acoustic prosodic features. We use the loudness and the Hammarberg index for voice quality, averaged over short-time frames.

Hidden-Unit BERT (HuBERT) (Hsu et al., 2021) extracts frame-level embeddings encapsulating phonetic, prosodic, and speaker-specific features via self-supervised learning. Normalized pitch, voice quality, and loudness scores are concatenated with HuBERT embeddings to form enhanced audio tokens $\{a_1, a_2, \ldots, a_p\}$, capturing dynamics such as emphatic speech. The token sequence, with SOS, EOS, and positional encodings, is processed by an audio encoder:

$$\{\hat{a}_i\}_{i=0}^{p+1} = \textbf{A-Encoder}\big(\{\texttt{SOS}, a_1, \ldots, a_p, \texttt{EOS}\}\big) \tag{2}$$

These representations encapsulate both phonetic content and spectro-temporal characteristics but require additional processing to capture the contextual relationships that characterize prosodic phenomena such as intonational patterns, rhythmic structures, and paralinguistic cues.

**Text processing.** Given a transcript as a sequence of sentences extracted from the audio, this pipeline extracts semantic embeddings and contextualizes them for multimodal fusion. We employ a state-of-the-art pretrained sentence-based language model (Reimers & Gurevych, 2019; Liu et al., 2019) to derive linguistic embeddings of the raw text. To facilitate discourse-aware learning for video summarization, we process these embeddings through a text encoder (T-Encoder), comprising a stack of transformer encoder layers. Augmented with start-of-sequence (SOS) and end-of-sequence (EOS) tokens and positional encodings, the sequence undergoes multi-head self-attention to model thematic flow:

$$\{\hat{s}_i\}_{i=0}^{k+1} = \textbf{T-Encoder}\big(\{\texttt{SOS}, s_1, \ldots, s_k, \texttt{EOS}\}\big) \tag{3}$$

The text encoder outputs contextualized embeddings, mean-pooled into a vector for cross-modal fusion.

**Cross-modal attention.** The cross-modal attention mechanism integrates visual, text, and audio modalities. Given encoded visual features $\hat{V} \in \mathbb{R}^{n \times d}$, text features $S \in \mathbb{R}^{k \times d}$, and audio features $A \in \mathbb{R}^{p \times d}$, we normalize and concatenate text and audio features to form $C = [S; A] \in \mathbb{R}^{(k+p) \times d}$. Visual features serve as queries ($Q = \hat{V} W^Q$), while text-audio features serve as keys ($K = C W^K$) and values ($V = C W^V$):

$$\text{Attention}(Q, K, V) = \text{softmax}\left(\frac{QK^T}{\sqrt{d_k}}\right)V \tag{4}$$

This implementation is followed by residual connections and a feed-forward network with layer normalization, allowing dynamic weighting of cross-modal relationships while preserving modality-specific characteristics. The cross-modal attention module fuses information across modalities, producing context-rich multimodal features by conditioning visual content on both textual and acoustic information. These features are subsequently utilized as context in the decoder network to generate the video summary.

**Summary generation.** Given encoded multimodal features, this pipeline decodes summary moments, and maps them to video segments for summarization. Temporal embeddings are added to input sequences using a positional encoding mechanism, ensuring the model captures sequential order. These embeddings, precomputed for a maximum length of 5000, are added to the input sequences to preserve temporal context. The generation process employs a transformer-based summary decoder (Vaswani et al., 2017). The decoder uses multimodal embeddings from cross-modal attention as context and a target sequence initialized with a start-of-sequence (SOS) token to predict the next summary frame. Positional encodings are applied to the target sequence, followed by decoding with a square subsequent mask to ensure autoregressive generation:

$$\hat{f}_t = \textbf{Decoder}\big(\{\text{multimodal}\}, \{\texttt{SOS}, f_1, \ldots, f_{t-1}\}, \text{mask}\big) \tag{5}$$

In the evaluation phase, the decoder begins with SOS token and iteratively constructs the sequence of key frames, incorporating previous outputs as input, until the EOS token is decoded. The resulting sequence is mapped to video segments using cosine similarity between decoded embeddings and CLIP-derived visual features of the input video, selecting indices that highlight significant interview moments for a concise summary.

## 3.2 PSEUDO-GROUND TRUTH SUMMARY GENERATION

With human-annotated summaries unavailable for the ChaLearn dataset, we develop a two-stage approach to generate pseudo-ground truth references for single-speaker interview videos. First, we employ heuristic-based methods to detect significant behavioural changes in facial expressions, prosodic patterns, and gestural emphasis using traditional computer vision and signal processing techniques (detailed in Section A.3 of the Appendix). These behavioural markers are then integrated with timestamped transcripts as metadata to guide LLMs in generating extractive summaries that prioritize segments exhibiting both semantic importance and behavioural salience. This LLM-driven approach (Narasimhan et al., 2022; Argaw et al., 2024; Zhang et al., 2024) provides an automated and scalable solution for behaviourally-informed reference summary generation.

**Task:** Generate an extractive summary from a timestamped video transcript that prioritizes sentences with both high semantic importance and behavioural salience.
**Guidelines:** 1) Select sentences aligned with behavioural cue timestamps, 2) Preserve exact wording, 3) Maintain original timestamps, 4) Output as [start_time, end_time, sentence].
**Input:** Transcript entries [start_time, end_time, sentence] with behavioural cue annotations [timestamp, cue_type] for facial movements, emotional transitions, pitch variations, prosodic emphasis, and voice quality shifts.
**Output:** Extractive summary as [start_time, end_time, sentence] triplets representing behaviourally-salient segments.

The process begins with transcribing the audio and aligning text with time markers using Whisper (Radford et al., 2023) and MFA (McAuliffe et al., 2017) to ensure precise correspondence between spoken content and video frames. We employ GPT-4.5 (Achiam et al., 2023) with tailored prompts that incorporate both the timestamped transcript and behavioural annotations (detected through the heuristic method) to perform extractive summarization, selecting key excerpts based on combined semantic importance and behavioural salience. The selected text segments are then mapped to corresponding video segments using their timestamps, converted to frame ranges, and concatenated chronologically to form a cohesive pseudo-ground truth video summary that maintains temporal alignment with the spoken content and behavioural markers.

## 4    EXPERIMENTS

We describe the experimental setup and evaluation results for the proposed behaviour-aware multimodal video summarization framework. Our approach is benchmarked against state-of-the-art methods using a combination of text and video metrics to evaluate performance.

**Evaluation Metrics.**   We evaluate our multimodal summarization framework on the ChaLearn First Impressions dataset, focusing on interview-specific summaries compared against pseudo-ground truth references. To assess summary quality, we follow prior text and video summarization approaches (Rochan et al., 2018; Otani et al., 2019; Islam et al., 2024) and employ a comprehensive set of text and video metrics. Text-based metrics include ROUGE-N for n-gram overlap and ROUGE-S phrase matching (Lin, 2004), BLEU (Papineni et al., 2002) for precision, and BERTScore (Zhang et al., 2019a) for semantic similarity. Length ratio measures summary brevity relative to the full transcript. For video-based evaluation, we assess the alignment and temporal consistency of the model-generated summaries against the pseudo-ground truth summary video as reference using F1-score, Kendall's $\tau$ (Kendall, 1945), Spearman's $\rho$ (Zwillinger & Kokoska, 1999), and CLIPScore (Hessel et al., 2021). Together, these metrics provide a comprehensive evaluation of segment relevance and temporal structure preservation.

### 4.1    EXPERIMENTAL RESULTS

We evaluate our method against existing state-of-the-art video summarization approaches, including CLIP-It (Narasimhan et al., 2021), MFST (Park et al., 2022) and Argaw et al. (2024) on the ChaLearn dataset, focusing on interview-based summaries. To ensure fair comparison, we adhere to their implementations and reimplement them as their official source code is unavailable, adapting them to our dataset's context. For CLIP-It, we adapt its approach by scoring frames based on cosine similarity between CLIP embeddings of input frames and a set of summary frames, computing the highest similarity score to determine frame priority. For Argaw et al. (2024), we reimplement their approach by encoding video frames and transcriptions with CLIP and SRoBERTa, respectively, using a transformer-based network to autoregressively generate interview-focused summaries. While more recent approaches (Zhang et al., 2023; Fajtl et al., 2019; Guo et al., 2025) show promising results in generalized video summarization techniques, they were excluded due to their limited applicability to dialogue-heavy interviews and their focus on visual diversity over semantic or prosodic content. Our selected baselines represent established multimodal summarization benchmarks balancing visual and textual information (Narasimhan et al., 2021; Argaw et al., 2024) with audio integration (Park et al., 2022), excelling in cross-modal scenarios (Jangra et al., 2023; Hua et al., 2025) that align with our behaviour-aware evaluation framework.

Table 1 highlights the evaluation of text-based summaries, where our approach achieves a significant improvement over the baselines, with a BLEU score of 0.4166 (45.1% improvement over Argaw et al. (2024)), and a BERTScore of 0.9247, indicating enhanced semantic fidelity compared to Argaw et al. (2024) and CLIP-It. While CFSum (Xiao et al., 2023) enhances ROUGE metrics over CLIP-It via coarse-to-fine multimodal fusion, it underperforms our framework in capturing behavioural nuances, yielding lower BLEU and BERTScore. ROUGE scores across all variants further confirm our framework's ability to generate behaviour-aware and contextual summaries, outperforming the baselines by at least 18% in ROUGE-1.

Table 1: **Performance comparison of text-based video summarization approaches on the ChaLearn dataset.** Our multimodal approach demonstrates significant improvements across all metrics compared to SOTA methods.

| Method | Length Ratio | ROUGE-1 | ROUGE-2 | ROUGE-L | ROUGE-S | BLEU | BERTScore |
|---|---|---|---|---|---|---|---|
| CLIP-It (Narasimhan et al., 2021) | 0.3785 | 0.4935 | 0.4120 | 0.4667 | 0.3800 | 0.2139 | 0.8984 |
| CFSum (Xiao et al., 2023) | - | 0.5723 | 0.4621 | 0.5841 | - | 0.3928 | 0.8977 |
| Argaw et al. (2024) | 0.4203 | 0.5529 | 0.4910 | 0.5333 | 0.4515 | 0.2871 | 0.9113 |
| Ours (Multimodal) | **0.6011** | **0.6765** | **0.6086** | **0.6442** | **0.5531** | **0.4166** | **0.9247** |

Our proposed framework significantly outperforms existing state-of-the-art methods across all video-based evaluation metrics, as shown in Table 2. The comprehensive comparison includes CLIP-It, which relies primarily on vision-based scoring, approach by Argaw et al. (2024) utilizing visual and textual features; and MFST (Park et al., 2022), which incorporates multimodal features but with a different architectural approach.

Table 2: **Quantitative evaluation of video-based summarization approaches on the ChaLearn dataset.** Our multimodal framework outperforms existing methods across all metrics. F1-Score, Kendall's $\tau$ and Spearman's $\rho$ highlight our model's superior ability to preserve the narrative flow of the original video, while CLIPScore demonstrates better visual-semantic alignment.

| Method | F1-Score | Kendall's $\tau$ | Spearman's $\rho$ | CLIPScore |
|---|---|---|---|---|
| CLIP-It (Narasimhan et al., 2021) | 0.6087 | 0.5949 | 0.5950 | 0.4918 |
| Argaw et al. (2024) | 0.7559 | 0.6359 | 0.6361 | 0.4827 |
| MFST (Park et al., 2022) | 0.7272 | 0.4500 | 0.6029 | 0.4970 |
| Ours (Multimodal) | **0.8107** | **0.6473** | **0.6466** | **0.5173** |

The MFST method demonstrates strong frame selection capabilities with an F1-Score of 0.7272, but its relatively low Kendall score reveals significant limitations in preserving temporal consistency compared to other approaches. Our multimodal framework substantially outperforms all baseline methods, achieving an F1-Score of 0.8107 (7.3% and 33.2% gain over Argaw et al. (2024) and CLIP-It, respectively). Temporal consistency metrics further underscore this advantage, with our model attaining Kendall's $\tau$ of 0.6765 and Spearman's $\rho$ of 0.6086, surpassing SOTA approaches. Additionally, our framework improves visual-semantic alignment, as evidenced by CLIPScore of 0.5173, a 5.3% increase over CLIP-It.

The effectiveness of our approach derives from the integration of three complementary modalities, where our framework distinctively incorporates vocal inflections and speech patterns from interviews that enhance the contextual understanding of visual scenes and transcribed texts. While MFST wasn't evaluated using text-based metrics due to its architecture focusing solely on frame importance scoring without text generation capabilities, our comprehensive evaluation demonstrates the clear advantages of our multimodal approach. By implementing an adaptive attention mechanism that balances modal influences based on context-specific needs, we generate more coherent and semantically rich summaries. Our decoding strategy further enhances quality by conditioning each summary moment on prior outputs, improving sequential coherence throughout the interview narrative. Though our consistency metrics remain moderate ($< 0.7$), suggesting opportunities for further refinement in capturing narrative structures, the comprehensive improvements across all measures validate our multimodal approach for interview summarization. Please see Section A.1 and A.3 of the Appendix for more details.

## 4.2 ABLATION STUDIES

In Table 3, we conduct an ablation study to assess the contributions of various components in our proposed framework on the ChaLearn dataset. We first explore the impact of excluding textual information, relying solely on audio-visual inputs. The configuration produces acceptable performance, but incorporating textual data significantly improves summarization quality, highlighting the strength of multimodal integration over an audio or visual-only approach. Similarly, omitting audio features while retaining text and visual inputs yields reasonable outcomes, yet adding audio provides additional context, emphasizing its supplementary role in capturing subtle interview dynamics. Removing visual features, however, leads to a marked decline in effectiveness, underscoring the foundational role of visual data in interpreting video content and structure.

Table 3: **Ablation study results for text-based and video-based metrics on the ChaLearn dataset.** 'R-' stands for ROUGE- metric, 'Ken' for Kendall, 'Spea' for Spearman coefficient, 'BS' for BERTScore, and 'CS' for CLIPScore.

| Method | Text-based Metrics | | | | | | Video-based Metrics | | | |
|---|---|---|---|---|---|---|---|---|---|---|
| | R-1 | R-2 | R-L | R-S | BLEU | BS | F1-Score | Ken's $\tau$ | Spea's $\rho$ | CS |
| Video Only | 0.5948 | 0.5385 | 0.5765 | 0.5021 | 0.3581 | 0.8810 | 0.7986 | 0.6281 | 0.6265 | 0.4761 |
| Text Only | 0.6341 | 0.5746 | 0.6130 | 0.5339 | 0.3596 | 0.8804 | 0.7603 | 0.6265 | 0.6283 | 0.4761 |
| Audio Only | 0.4390 | 0.3923 | 0.4217 | 0.3558 | 0.2139 | 0.7276 | 0.6460 | 0.5207 | 0.5174 | 0.3924 |
| w/o Audio | 0.6329 | 0.5748 | 0.6121 | 0.5337 | 0.3585 | 0.8806 | 0.7975 | 0.6287 | 0.6271 | 0.4748 |
| w/o Text | 0.6308 | 0.5753 | 0.6098 | 0.5333 | 0.3590 | 0.8794 | 0.7986 | 0.6247 | 0.6268 | 0.4747 |
| w/o Video | 0.5322 | 0.4737 | 0.5087 | 0.4315 | 0.2582 | 0.8813 | 0.6986 | 0.6278 | 0.6270 | 0.4776 |
| w/o CM-Attn | 0.6267 | 0.5693 | 0.6056 | 0.5280 | 0.3568 | 0.8729 | 0.6947 | 0.6199 | 0.6208 | 0.4712 |
| **Proposed** | **0.6765** | **0.6086** | **0.6442** | **0.5531** | **0.4166** | **0.9247** | **0.8107** | **0.6473** | **0.6466** | **0.5173** |

Further evaluation examines the contribution of cross-modal learning. excluding cross-attention mechanism (CM-Attn) exhibits a noticeable drop in performance, indicating that cross-modal interactions are essential for effectively combining all the modalities. Assessing each modality independently reveals that visual features offer the strongest stand-alone performance, followed by text, with audio being the least effective, yet all single-modality fall short compared to multimodal configurations, reinforcing the value of integration. Our full multimodal framework, which integrates visual, textual, and audio features through adaptive cross-modal learning and autoregressive decoding, consistently achieves the highest performance across most metrics. The superior results in all the metrics confirm that combining all modalities enhances summarization quality, affirming our hypothesis that the multimodal approach significantly improves behaviour-aware video summarization in interview-focused scenarios. Please see Section A.4 of the Appendix for additional ablations.

## 5 CONCLUSION

This paper introduces a behaviour-aware multimodal video summarization framework that advances the state-of-the-art by integrating visual, audio, and textual modalities using cross-modal attention mechanisms. Our approach captures synchronized behavioural features: CLIP embeddings with facial movements and emotional transitions, HuBERT audio representations with prosodic patterns, and contextualized text embeddings to convey communicative intent in interview videos. Addressing the absence of annotated data, we develop a heuristic-based pseudo-ground truth generation technique guided by detected behavioural cues using LLMs. Experiments on the ChaLearn First Impressions dataset show significant metric improvements, with ablation studies confirming that cross-modal attention optimizes performance, setting a new benchmark in behaviour-aware summarization.

Our framework's modular design offers inherent advantages for domain generalization through interpretable, domain-agnostic feature extraction and threshold calibration, facilitating transfer learning across content domains by separating behavioural cue detection from deep learning components. Comprehensive analysis, including case studies and failure modes (Appendix Section A.5-A.8 of the Appendix), highlights its strength in detecting synchronized behavioural emphasis, though multi-speaker and diverse contexts warrant further exploration. Future work will focus on cross-domain validation across various video types and human evaluation studies to strengthen the link between detected cues and perceived importance, laying a foundation for behaviour-aware video understanding with broad implications for educational technology, accessibility, and human-centered AI.

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

# A  APPENDIX

## A.1  SUPPLEMENTARY IMPLEMENTATION SPECIFICATIONS

**Data preprocessing.**  Our preprocessing step isolates and synchronizes visual, audio, and textual streams for multimodal analysis. Frames are sampled at 1 fps to capture key visual moments (e.g., expressive gestures). Audio streams are also extracted using FFmpeg, providing high-quality and single-channel audio files (.wav) at 16 kHz sampling rate. The Whisper automatic speech recognition (ASR) model (Radford et al., 2023) transcribes audio, handling diverse speech conditions such as, accents, minor background noise with high accuracy. Whisper produces a raw transcript, segmenting the audio into sentences or phrases based on pauses and intonation. Transcripts are then structured using SpaCy (Honnibal et al., 2020), which performs sentence boundary detection to correct run-on sentences, adds punctuation (e.g., periods, commas), and normalizes text by converting to lowercase and removing extra whitespace. The Montreal Forced Aligner (MFA) (McAuliffe et al., 2017) aligns each transcribed word with its precise audio timestamp using hidden Markov models and Kaldi-based pretrained English acoustic models, enabling millisecond-level synchronization with video frames and text. MFA matches the audio's phonetic features to the text, generating start and end timestamps for each word. Sentence-level timestamps are derived by aggregating word timestamps. Alignment accuracy is verified to ensure sentence timestamps correspond to video frames (e.g., 7.21s to 8.51s maps to frames 173–204 at 24 fps), with a 50ms tolerance for minor discrepancies. MFA's precision, robustness, and granularity make it the optimal choice for accurate temporal alignment in this framework, outperforming other alignment methods for cross-modal synchronization.

**Implementation details.**  The multimodal summarization is optimized for the ChaLearn First Impressions dataset (Ponce-López et al., 2016), utilizing its controlled single-speaker interview setting. Visual features are extracted at 1 fps, forming a sequence of frames for both input videos and pseudo-ground truth summaries. Feature encoding utilizes CLIP-ViT-large-patch14[1] (Radford et al., 2021) for visual embeddings, HuBERT-base-ls960[2] (Hsu et al., 2021) for audio embeddings (16 kHz sampling rate, 20 ms frames), and RoBERTa-large[3] (Liu et al., 2019) for text embeddings. The architecture comprises a video encoder, a text encoder and an audio encoder, a cross-modal attention mechanism, and a summary decoder. Each layer has a transformer-based architecture with 6 layers, 8 attention heads, and a 2048-dimensional feed-forward network, incorporating dropout at 0.1. Decoding initiates with a start-of-sequence (`SOS`) token and proceeds iteratively, generating summary frames using a subsequent mask to enforce sequential dependency. Final summary alignment maps decoded frames to video segments by computing cosine similarity between 1024-dimensional embeddings and CLIP-derived visual features of the input video. Selected indices, corresponding to significant interview moments, are converted to timestamps, ensuring a concise and representative output.

**Multimodal cues detection.**  Behavioural emphasis for visual cues in our framework is derived from two key sources: head movement trajectories, as illustrated in Figure 2, facial emotion transitions, and semantic visual understanding. To ensure consistent analysis, we extract frames at fixed intervals using OpenCV Bradski (2000). We select the nose landmark to represent head position. Any displacement exceeding the threshold ($\lambda$) was flagged as a head movement cue, indicating physical emphasis or non-verbal communication. For facial emotions, we analyze each frame and a visual cue is recorded when a transition occurs between distinct emotions. These traditional features are complemented with CLIP Radford et al. (2021) visual embeddings (512-dimensional), which capture high-level semantic content through its Vision Transformer (ViT) architecture pretrained on 400M image-text pairs, enabling our model to recognize contextually significant visual elements beyond simple motion or emotion detection.

Next, we process audio streams to detect prosodic emphasis through three key acoustic features: pitch, loudness and voice quality, enriched with HuBERT Hsu et al. (2021) embeddings. HuBERT leverages self-supervised learning on 960 hours of speech data to extract frame-level representations that capture phonetic, prosodic, and speaker-specific characteristics. Pitch ($F_0$) is extracted to handle

---

[1] openai/clip-vit-large-patch14

[2] facebook/hubert-base-ls960

[3] sentence-transformers/all-roberta-large-v1

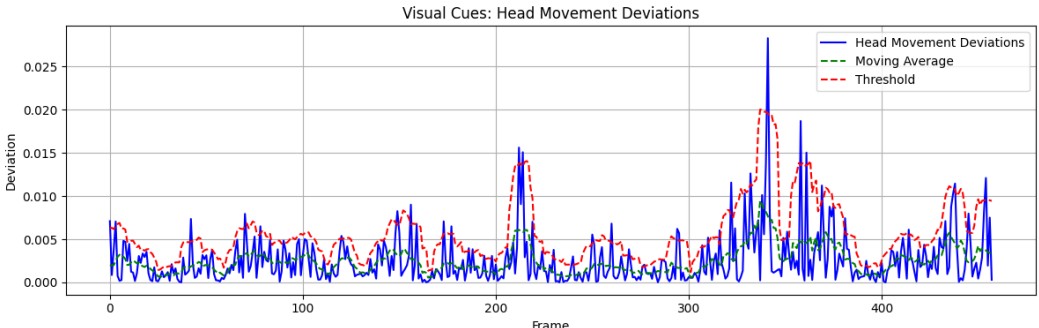

Figure 2: **Example of head movement detection for behavioural visual cue identification.** The graph displays frame-by-frame head position deviations (blue line) measured as Euclidean distance between consecutive frames using facial landmark tracking. The moving average (green dashed line) smooths the signal, while the adaptive threshold (red dashed line) identifies significant movements. Notable spikes around frames 200, 325, and 400 correspond to expressive head gestures that likely indicate moments of emphasis or emotional significance during the interview, which our framework leverages as behavioural cues for summary generation.

noise and unvoiced segments, loudness is quantified via short-time root mean square (RMS) energy and voice quality is assessed using the $dB$ difference between the strongest harmonic peak in 0–2 kHz and 2–5 kHz ranges of the speech spectrum. This index characterizes spectral slope, with lower values indicating flatter spectra (suggesting vocal strain) and higher values reflecting greater low-frequency energy (associated with breathier voice). Figure 3 provides an example of pitch ($F_0$), loudness, and voice quality (Hammarberg Index) variations over time.

**Summary video compilation.** The decoded sequences from our proposed framework undergo systematic post-processing to synthesize the final summary video. We first transform the identified key moments into precise temporal frame boundaries by mapping each predicted segment to specific frame indices using the source video's native frame rate. This temporal alignment ensures frame-accurate extraction while preserving the semantic integrity of selected content. For segment extraction, we leverage FFmpeg's advanced filtering capabilities with optimized parameters to preserve perceptual quality during extraction. We synchronize textual content with visual segments and each sentence from the transcript is mapped to its corresponding time interval using our millisecond-precision alignment data generated during preprocessing. The subtitle integration employs a custom rendering pipeline to ensure readability across diverse viewing conditions. The extracted segments undergo temporal concatenation that preserves frame continuity and audio transitions while maintaining codec consistency. This process yields a cohesively structured summary that effectively condenses the original content while retaining the multimodal behavioural cues critical for understanding the speaker's communicative intent. The resulting output represents approximately 60% of the original duration, representing a balance between conciseness and comprehensive coverage of semantically salient content.

## A.2 GENERALIZATION TO SMALLER MODELS

LLMs are provided identical prompts requesting extractive summaries of transcripts with timestamp preservation. The results in Table 4 reveal that our framework achieves optimal performance when using GPT-4.5 for pseudo-ground truth summary generation. Summaries generated by GPT-4.5 attain the highest F1-Score compared to GPT-3.5 and LLaMA-3.2. Our analysis on the LLM-generated summaries indicate that GPT models produce more balanced summaries and preserve contextual information, whereas 3B variant of LLaMA-3.2 model exhibits less consistent adherence to the extractive constraints specified to our prompts and sometimes retained less significant content or produce redundant content. These findings highlight the importance of selecting the most capable language model for generating high-quality pseudo-ground truth summaries in multimodal summarization evaluation.

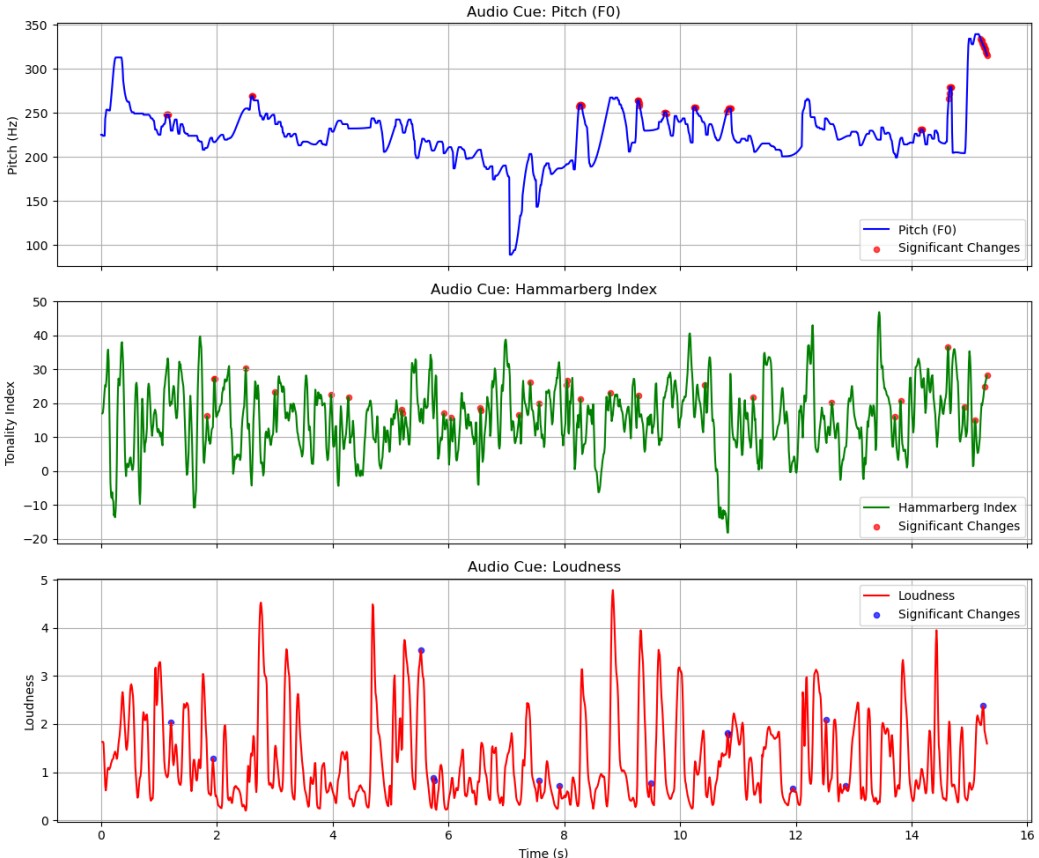

Figure 3: **Example of prosodic features for audio cue detection.** The figure shows three acoustic features extracted from an interview audio: (top) fundamental frequency/pitch ($F_0$) tracking intonation patterns; (middle) Hammarberg Index measuring voice quality (spectral slope); and (bottom) loudness measurements capturing speech intensity, with significant audio cues marked as round.

Table 4: **Impact of different LLMs for pseudo-ground truth summary generation.** Our evaluation compares the quality of summaries generated by three large language models when used as reference for evaluation. GPT-4.5 consistently produces the highest quality reference summaries, leading to better metric scores across all evaluation dimensions, while GPT-3.5 and LLaMA-3.2 show progressively lower performance.

| Method | F1-Score | Kendall's $\tau$ | Spearman's $\rho$ | CLIPScore |
|---|---|---|---|---|
| LLaMA-3.2 | 0.6823 | 0.5417 | 0.5213 | 0.4125 |
| GPT-3.5 | 0.7965 | 0.6358 | 0.6342 | 0.5027 |
| GPT-4.5 | **0.8107** | **0.6473** | **0.6466** | **0.5173** |

While these findings highlight the importance of selecting the most capable language model for generating high-quality references, we acknowledge that reliance solely on pseudo-ground truth summaries raises concerns regarding evaluation reliability. To address this, we implement several methodological safeguards. First, our summary generation process using LLMs employs a carefully designed prompt-engineering approach that constrains the LLM to perform extractive summarization only, preserving the exact wording and structure of original sentences. This eliminates potential hallucination issues that might arise with abstractive approaches and maintains fidelity to the source content. Then, we implement a consistency verification procedure where we generate three independent summaries for a subset of randomly selected videos using different LLM parameters. The high inter-summary agreement (average Jaccard similarity of 0.83) demonstrates the stability of our

LLM-based summary generation approach. The analysis revealed that about 93% of summaries maintained high coverage of key information points, with minimal redundancy and strong temporal coherence.

### A.3 EXPERIMENTAL ANALYSES

**Classification approach.** We investigate our approach with the traditional binary classification framework. In our framework, summary generation follows a decoding strategy where each prediction conditions on previously selected segments. For comparison, we implement a binary classification alternative that replaces our temporal decoder with a frame-level binary classifier using a fully-connected layer with sigmoid activation against the same reference summaries.

Our experimental results, presented in Table 5, demonstrate the significant advantages of the sequential approach. The binary classification model achieves moderate F1-Score, yet lower than our proposed model. Similar performance gaps exist across temporal consistency metrics and semantic alignment measures. These findings suggest that frame-by-frame classification, while computationally simpler, fails to capture the crucial narrative dependencies between summary moments. The sequential model's strength lies in its ability to model temporal relationships through iterative conditioning, producing more coherent summaries that maintain narrative integrity.

Table 5: **Comparison between our proposed and binary classification approaches for video summarization.** Our proposed model outperforms the binary classification approach, demonstrating the advantages of modeling temporal dependencies through decoding rather than independent frame-level decisions.

| Method | F1-Score | Kendall's $\tau$ | Spearman's $\rho$ | CLIPScore |
|---|---|---|---|---|
| Binary Classification | 0.7214 | 0.5479 | 0.5295 | 0.4826 |
| Ours (proposed) | **0.8107** | **0.6473** | **0.6466** | **0.5173** |

**Multimodal heuristic approach.** While our main manuscript focuses on the transformer-based architecture with cross-modal attention for behavioural feature fusion, this approach provides an entirely separate, interpretable, and rule-based method that serves two critical functions: (1) identifying behaviourally significant moments for pseudo-ground truth reference generation using LLMs, and (2) generating video summaries as a standalone alternative to the transformer-based decoder architecture. This approach prioritizes key terms identified as bonus words (a concept popularized in Edmundson's summarizer (Edmundson, 1969)) that temporally align with significant visual (e.g., pose shifts, emotional changes), textual, and audio (e.g., pitch peaks, loudness variations) cues. These bonus words are weighted using their frequency and multimodal relevance. For example, when a speaker emphasizes a point through simultaneous gesturing and vocal stress, this approach captures this cross-modal emphasis with precise frame timestamps, which then serves dual purposes: informing LLM prompts for behaviourally-aware pseudo-ground truth generation and directly contributing to heuristic-based summary selection.

To implement this dual-purpose approach, we apply a fundamentally different importance scoring mechanism that operates independently of deep learning architectures. First, we assign weights to sentences based on their detected behavioural bonus words, which increases their likelihood of selection in both the LLM-guided reference summaries and the direct heuristic summarization process. Second, we prioritize video segments with higher bonus word density during segment selection, ensuring that moments with rich multimodal cues are preserved in both evaluation references and heuristic-generated summaries. Third, we apply a diversity-promoting filtering algorithm based on TF-IDF vectors and cosine similarity to prevent redundancy in the selected segments. Unlike our main transformer approach, this heuristic method does not utilize any form of autoregressive decoding or cross-attention mechanisms, instead relying entirely on rule-based sentence scoring and selection algorithms, following Edmundson's extractive summarization principles (Edmundson, 1969), while simultaneously providing the critical behavioural metadata that guides LLM-based pseudo-ground truth generation.

The implementation uses adaptive thresholds for detecting cross-modal emphasis across different modalities, creating a robust foundation for both summarization approaches. For head movement

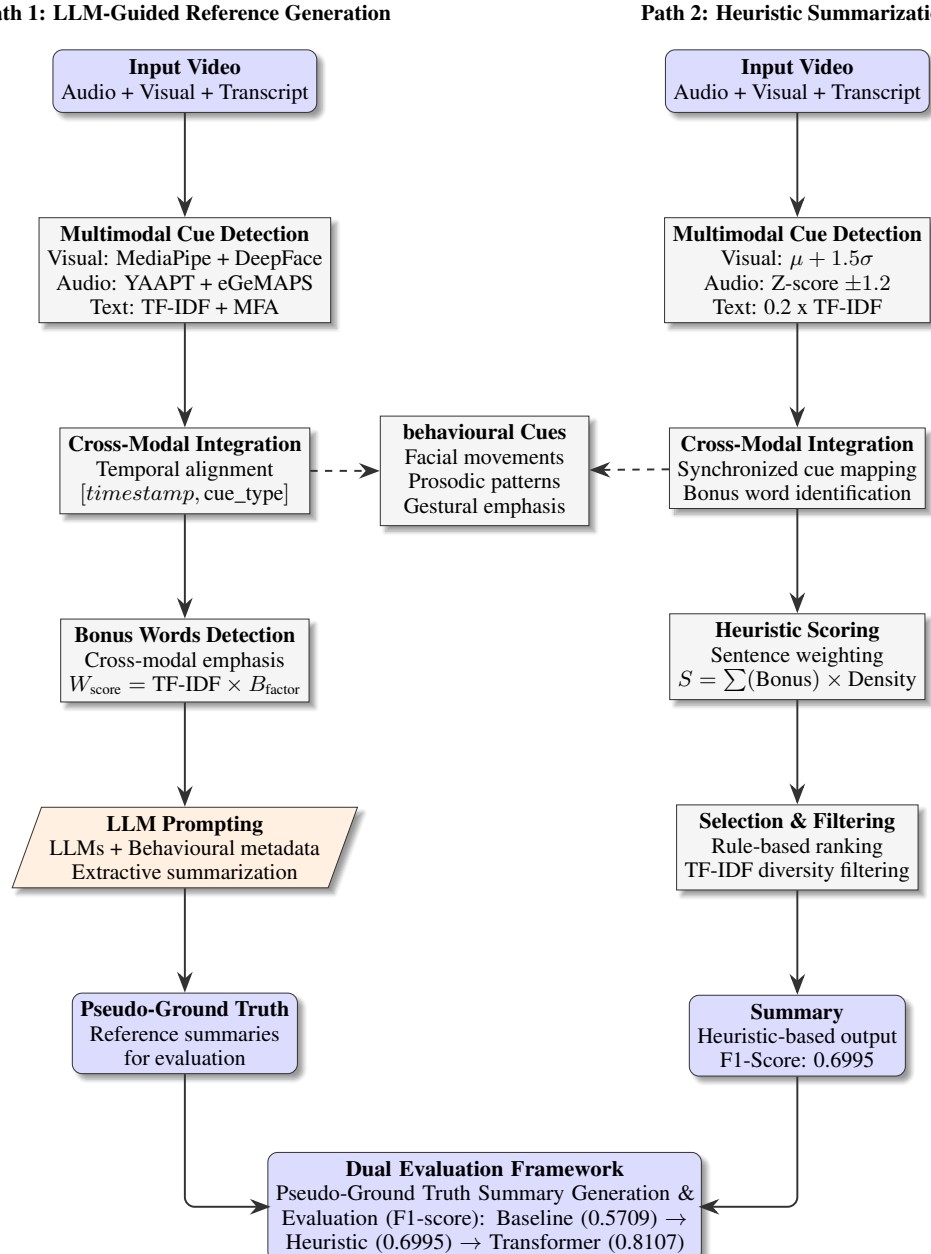

Figure 4: **Multimodal heuristic framework for behavioural cue detection and video summarization.** The framework operates through two parallel paths: (Left) LLM-guided pseudo-ground truth generation using detected behavioural metadata for evaluation, and (Right) extractive summarization using rule-based scoring mechanisms. Both paths share a common multimodal cue detection foundation but serve complementary functions.

detection, we flag significant movements when Euclidean displacement exceeds the mean plus 1.5 standard deviations, capturing deliberate gestures while filtering out minor involuntary movements. These detected movements are timestamped and used both as direct heuristic cues and as behavioural annotations in LLM prompts. Pitch variation is identified when Z-score normalized changes exceed $\pm 1.2$, highlighting vocal emphasis patterns that indicate communicative intent. Voice quality changes are detected when the Hammarberg Index, which characterizes spectral slope and vocal effort, shows fluctuations exceeding $\pm 2.0$ standard deviations. For textual significance, we consider terms in the top 20% by TF-IDF weight, focusing on content-rich words rather than functional terms. All detected behavioural cues are mapped to their corresponding transcript segments, creating enriched annotations that inform both the heuristic scoring mechanism and the LLM-based reference generation process.

Table 6: **Comparative evaluation of video summarization approaches.** Our proposed transformer-based architecture demonstrates superior performance across all metrics, while the heuristic approach also shows significant improvement over the Edmundson baseline.

| Method | F1-Score | Kendall's $\tau$ | Spearman's $\rho$ | CLIPScore |
|---|---|---|---|---|
| Edmundson | 0.5709 | 0.2295 | 0.2681 | 0.3513 |
| Ours (heuristic) | 0.6995 | 0.3148 | 0.3690 | 0.4162 |
| Ours (proposed) | **0.8107** | **0.6473** | **0.6466** | **0.5173** |

We evaluate this heuristic approach in both capacities: as a standalone video summarization method and as the behavioural cue detection foundation for our pseudo-ground truth generation pipeline. The results, presented in Table 6, demonstrate the effectiveness of this dual-purpose methodology. As a direct summarization approach, while it does not achieve the state-of-the-art performance of our proposed transformer model, this heuristic method demonstrates considerable improvement over traditional baselines with an F1-score of 0.6995 compared to Edmundson's baseline of 0.5709. More critically, the behavioural cues detected by this heuristic approach serve as the essential metadata that enables LLMs to generate behaviourally-informed pseudo-ground truth references. This creates a synergistic relationship where traditional computer vision and signal processing techniques inform modern language models, resulting in evaluation references that capture both semantic importance and behavioural salience.

The behavioural annotations generated through this heuristic approach are formatted as [timestamp, cue_type] triplets, where cue_type includes facial movement, emotional transitions, pitch variation, loudness change, or voice quality shift. These annotations are integrated with timestamped transcripts and provided to LLMs (GPT-4.5, GPT-3.5, LLaMA-3.2) as contextual metadata, guiding the extractive summarization process to prioritize segments that exhibit cross-modal behavioural emphasis. LLMs receive prompts that include both the raw transcript and these behavioural markers, enabling them to make informed decisions about which sentences to select based on combined semantic and behavioural criteria.

**Experimental setup.** The pipeline runs on a 32GB NVIDIA V100 GPU, requiring approximately 200GB of SSD storage for the 1,500 video dataset and intermediate files. Each experimental run on the full dataset takes about 10 GPU hours, with a total compute of approximately 50 GPU hours across the reported experiments. Preliminary experiments required an additional 20 GPU hours, though these are not detailed in the main results. For pseudo-ground truth summary generation, we utilize the GPT-4.5 and GPT-3.5 APIs to process transcripts from 1,500 videos, with an estimated 600 tokens per transcript (input and output combined). Based on the provided pricing for GPT models, the total cost is about $300.

### A.4 ADDITIONAL ABLATIONS

To complement the modality ablation studies in the main manuscript, we present additional experiments in Table 7 isolating the contributions of modality-specific encoders and feature enhancements on the ChaLearn First Impressions dataset.

**V-Encoder.** Removing the video encoder significantly impairs performance across all metrics, particularly affecting temporal consistency and behavioural detection. Without the video encoder's

Table 7: **Additional ablation study of encoder components and feature types.** Results show the impact of removing individual encoders and isolating specific features on both text and video-based metrics. The proposed method significantly outperforms all ablated variants. 'R-' stands for ROUGE-metric, 'Ken' for Kendall, 'Spea' for Spearman coefficient.

| Method | Text-based Metrics | | | | | | Video-based Metrics | | | |
|---|---|---|---|---|---|---|---|---|---|---|
| | R-1 | R-2 | R-L | R-S | BLEU | BERTScore | F1-Score | Ken's $\tau$ | Spea's $\rho$ | CLIPScore |
| w/o V-Encoder | 0.5226 | 0.4649 | 0.5024 | 0.4234 | 0.2539 | 0.8642 | 0.6710 | 0.6163 | 0.6141 | 0.4314 |
| w/o T-Encoder | 0.5019 | 0.4451 | 0.4791 | 0.4059 | 0.2425 | 0.8260 | 0.6970 | 0.6265 | 0.6280 | 0.4771 |
| w/o A-Encoder | 0.5255 | 0.4716 | 0.5069 | 0.4290 | 0.2561 | 0.8719 | 0.6955 | 0.6184 | 0.6182 | 0.4713 |
| CLIP Only | 0.5326 | 0.4770 | 0.5096 | 0.4328 | 0.2582 | 0.8790 | 0.6979 | 0.7268 | 0.6254 | 0.4962 |
| HuBERT Only | 0.5335 | 0.4739 | 0.5101 | 0.4327 | 0.2582 | 0.8809 | 0.6978 | 0.6280 | 0.6253 | 0.4789 |
| w/o CLIP | 0.5328 | 0.4740 | 0.5111 | 0.4314 | 0.2588 | 0.8802 | 0.5997 | 0.6265 | 0.6261 | 0.4759 |
| w/o HuBERT | 0.5317 | 0.4744 | 0.5122 | 0.4322 | 0.2585 | 0.8797 | 0.5980 | 0.6254 | 0.6275 | 0.4763 |
| **Proposed (Full)** | **0.6765** | **0.6086** | **0.6442** | **0.5531** | **0.4166** | **0.9247** | **0.8107** | **0.6473** | **0.6466** | **0.5173** |

self-attention mechanisms, the model struggles to capture sequential patterns in facial expressions and gestures that signal important moments. Raw CLIP embeddings provide semantic understanding but lack the temporal contextualization needed to identify behavioural significance in interview scenarios. This confirms the crucial role of the video encoder in establishing relationships between consecutive frames for coherent summarization.

**T-Encoder.** Our text encoder ablation demonstrates its importance for semantic alignment and narrative cohesion. Without the text encoder, the model relies on raw sentence embeddings from RoBERTa, missing discourse-level patterns and thematic progression within the transcript. This primarily affects text-based metrics, particularly BLEU and ROUGE scores, while maintaining reasonable performance on video-based metrics. The text encoder proves essential for identifying speech segments that complement visual behavioural cues.

**A-Encoder.** The audio encoder shows the smallest but still meaningful contribution among the three encoders. Its removal primarily affects the detection of prosodic emphasis and emotional voice modulation, which serve as complementary signals to visual cues in interview contexts. The relatively modest impact reflects the visually-dominant nature of the dataset, though audio remains valuable for detecting emphasis patterns not visible in facial expressions alone.

**Feature-specific ablations.** Our experiments isolating CLIP features from facial and emotional features reveal their complementary nature. CLIP-only configurations offer strong semantic understanding but miss fine-grained behavioural cues such as subtle head movements or emotional transitions. Conversely, using facial movements and emotional features captures behavioural dynamics but lacks broader semantic context, resulting in summaries that prioritize expressive moments without sufficient content relevance.

The contrast between HuBERT and prosodic features-based configurations demonstrates the value of contextualized speech embeddings over isolated acoustic features. HuBERT embeddings implicitly capture both linguistic content and paralinguistic cues, while explicit prosodic features, such as pitch, loudness, and voice quality, provide targeted detection of vocal emphasis but miss broader speech patterns. This explains why our full model benefits from incorporating both representation types for comprehensive audio understanding.

## A.5 CASE STUDY

To demonstrate the practical effectiveness of our behaviour-aware multimodal framework, we present a detailed case study using an example video from the ChaLearn First Impressions dataset. This analysis illustrates how detected behavioural cues guide LLM-based pseudo-ground truth generation.

**Video characteristics and behavioural detection.** The example video features a 15.3-second interview segment where a speaker discusses their motivation for writing popular history books. The complete transcript reads:

*"The short answer is that I really wanted to write this book because I absolutely love writing popular histories. I mean, I love the deep, detailed histories, of course. This is my thing. But I love being able to share these stories in a way that makes it really accessible and exciting for people that wouldn't necessarily"*.

Our heuristic detection pipeline identified several behavioural cues: facial movement peaks at 2.1s and 13.8s (deliberate head gestures), pitch variations at 4.2s and 8.9s (prosodic emphasis), and voice quality changes at 6.5s (increased vocal effort). These cues temporally align with semantically important content containing emotional expression ("absolutely love", "really accessible").

Figure 5 illustrates the temporal alignment of detected behavioural cues with the LLM selection process. The behavioural annotations are formatted as [timestamp, cue_type] and integrated into LLM prompts alongside the timestamped transcript.

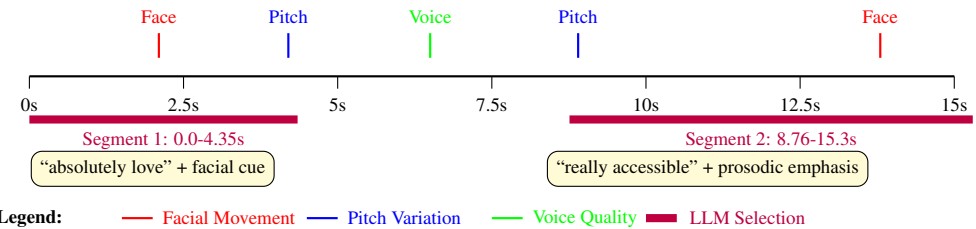

Figure 5: **Temporal alignment of behavioural cues and pseudo-ground truth summary selection for an example video.** The timeline shows detected facial movements (red), pitch variations (blue), and voice quality changes (green) aligned with LLM-selected summary segments (purple). Yellow boxes highlight key moments where cross-modal behavioural emphasis influenced extractive summarization decisions.

The LLM selected two segments: (1) 0.0-4.35s and (2) 8.76-15.3s, excluding the gap 4.35s-8.76s despite semantic relevance. This demonstrates how behavioural annotations guide LLMs to prioritize segments with stronger cross-modal emphasis, validating our dual-purpose heuristic methodology.

Table 8: **Behavioural cues detection and LLM-guided pseudo-ground truth generation process.** This table illustrates how heuristically detected behavioural cues inform LLM prompts to generate behaviourally-aware pseudo-ground truth summaries.

| Processing Stage | Detected Behavioural Cues | LLM-Selected Content |
|---|---|---|
| Behavioural Cue Detection | **Facial movements:** 2.1s, 13.8s (head gestures)
**Pitch variations:** 4.2s, 8.9s (prosodic emphasis)
**Voice quality:** 6.5s (vocal effort)
**TF-IDF words:** "absolutely," "really," "accessible" | **Segment 1 (0.0-4.35s):** "The short answer is that I really wanted to write this book because I absolutely love writing popular histories."
**Segment 2 (8.76-15.3s):** "But I love being able to share these stories in a way that makes it really accessible and exciting..." |
| LLM Integration | Behavioural annotations [timestamp, cue_type] provided in LLM prompt.
Example: [2.1s, facial_movement], [4.2s, pitch_variation], [6.5s, voice_quality], [8.9s, pitch_variation], [13.8s, facial_movement] | LLM prioritizes segments with behavioural emphasis while maintaining extractive integrity. The gap (4.35s-8.76s) is excluded despite semantic relevance due to the absence of cross-modal behavioural cues. |

The heuristic detection pipeline identifies behavioural emphasis moments that are then formatted as temporal annotations and integrated into LLM prompts. The LLM receives both the timestamped transcript and these behavioural markers, enabling it to make informed extractive summarization decisions that balance semantic importance with behavioural salience. This demonstrates how our detection pipeline supports our core methodology by providing behaviourally-informed pseudo-ground truth references for evaluation.

Figure 6 represents a visual comparison of summary outputs from different methods applied to an example. The figure illustrates how our behaviour-aware approach selects different temporal segments compared to baseline methods. Figure 7 provides a detailed temporal analysis of frame importance scores, comparing our proposed method against the pseudo-ground truth and baseline approaches.

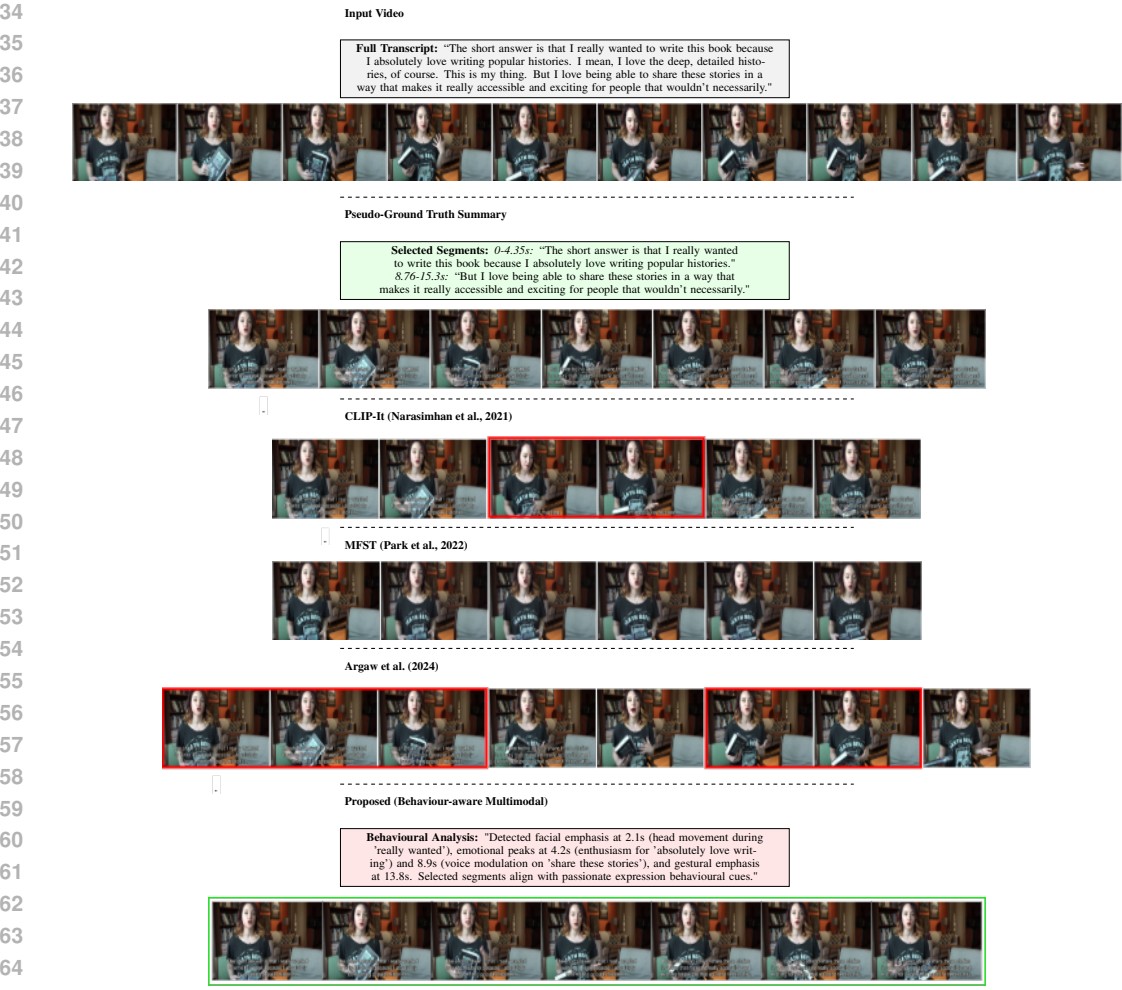

Figure 6: **Visual comparison of video summarization methods on an example video.** Each method's selected frames are shown with corresponding transcripts where applicable. Our behaviour-aware approach closely aligns with the LLM pseudo-ground truth by detecting facial expressions and prosodic patterns, while baseline methods show different selection patterns based on visual salience or multi-modal features.

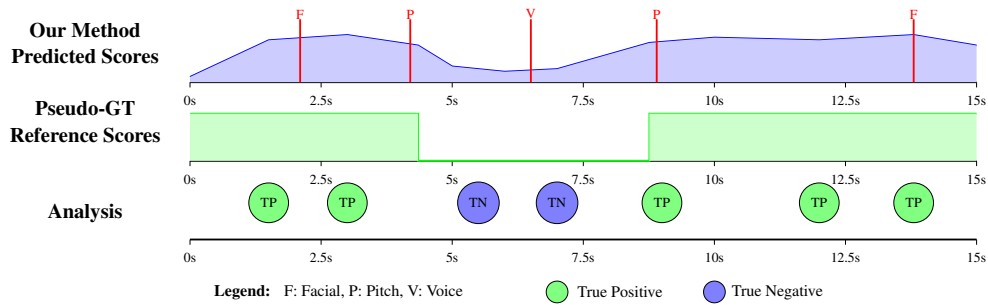

Figure 7: **Temporal score analysis for an example video.** Top: Our method's predicted importance scores showing peaks aligned with behavioural cues. Middle: Pseudo-ground truth binary scores indicating selected segments. Bottom: True positive/negative analysis showing our method's alignment with behaviourally significant moments. The high correlation demonstrates effective behavioural cue integration.

Our analysis reveals how behavioural cue detection supports the core transformer-based methodology outlined in Section 3 and validates its effectiveness through the generated summary output. The heuristic detection of facial movements at 2.1s and 13.8s provides temporal markers that enhance the visual processing pipeline's CLIP embeddings. These detected movements are concatenated with visual features as described in Equation 1, creating behaviourally-enriched visual tokens that inform the video encoder about moments of gestural emphasis. The prosodic variations detected at 4.2s and 8.9s serve as additional features in the audio processing pipeline, complementing HuBERT embeddings as specified in Equation 2. During cross-modal attention, these enriched audio features guide the attention mechanism to focus on temporally aligned visual and textual content, improving the integration described in the methodology.

The transformer-based decoder, trained using the LLM-generated pseudo-ground truth references, learned to identify and prioritize segments containing cross-modal behavioural emphasis. The output summary captures key moments at 0.0-4.35s (emotional emphasis on "absolutely love writing") and 8.76-15.3s (prosodic emphasis on "really accessible"), demonstrating that the cross-modal attention mechanism successfully integrated behavioural cues across modalities. Notably, the autoregressive decoding strategy maintained temporal coherence while the enhanced multimodal features enabled the model to distinguish between semantically relevant content and communicatively emphasized moments, resulting in a summary that preserves both narrative flow and behavioural salience.

The convergence between heuristically detected behavioural cues, LLM-selected segments, and the transformer-generated summary output validates our complete pipeline. This three-way alignment (behavioural cues detection → LLM segments → transformer output) demonstrates that our pipeline successfully identifies moments of genuine communicative intent, provides high-quality training references, and enables the transformer to learn behaviour-aware summarization patterns rather than spurious correlations.

## A.6 FAILURE CASES

Our behaviour-aware multimodal summarization framework generally performs well but exhibits specific failure cases worth noting. Since we rely on our decoder to determine summary length rather than using a fixed percentage threshold, some summaries mismatch reference lengths. For example, in shorter interview videos, our method sometimes generates summaries that are either too concise (missing contextual information) or too detailed (including less significant segments). Cross-modal attention occasionally over-prioritizes a single modality, particularly when visual features have high confidence scores, leading to summaries that miss semantically important content with subtle behavioural cues. Temporal misalignments between modalities also affect approximately 9% of cases despite our forced alignment approach, creating unnatural breaks or transitions in the summary.

## A.7 LIMITATIONS AND FUTURE WORK

**Limitations.** While our behaviour-aware multimodal framework significantly outperforms existing approaches, several limitations warrant attention. First, the model is optimized for single-speaker interview videos in controlled settings, such as those in the ChaLearn First Impressions dataset, limiting its applicability to multi-speaker scenarios or unstructured content. For instance, multi-speaker interactions involve overlapping speech and complex visual dynamics, such as tracking multiple faces or handling dynamic backgrounds, which our framework is not designed to address. Second, reliance on LLM-generated pseudo-ground truth summaries introduces biases, such as LLMs prioritizing semantically dense sentences over emotionally nuanced content with subtle behavioural cues, which may misalign with human preferences, particularly for short videos ($< 10$ seconds) where extractive summarization struggles. Additionally, the cross-modal attention mechanism occasionally over-weights visual features, missing semantically significant speech content, while temporal misalignments between modalities, due to transcription inaccuracies or alignment errors, affect approximately 9% of generated summaries.

**Future work.** To enhance our framework, we plan to incorporate more comprehensive behavioural representations beyond facial, prosodic, and textual features, potentially including eye gaze patterns and body posture analysis. We will explore integration with larger-scale pretrained vision-language models and audio foundation models to enhance feature representation quality. Developing adaptive

modality balancing techniques, such as dynamic attention weighting based on modality confidence scores, modality dropout to prevent overfitting to dominant modalities, or learnable modality fusion layers could optimize cross-modal integration. Additionally, expanding our framework to diverse video genres (e.g., multi-speaker discussions, vlogs) and datasets such as MultiSum (Qiu et al., 2023) would improve generalizability. Human evaluation studies, including subjective quality assessments and A/B testing, would provide insights into perceptual quality compared to LLM-generated references, guiding human-aligned optimization. Finally, we aim to address synchronization challenges by investigating robust temporal alignment methods, such as end-to-end audio-visual synchronization models, and advanced decoder architectures, such as memory-augmented or external knowledge-based transformers (Feng et al., 2018; Xie et al., 2022; He et al., 2024) to capture long-range dependencies or graph-based decoders to model inter-segment relationships, to improve summary coherence across varying video durations.

## A.8 BROADER IMPACT

Our behaviour-aware multimodal video summarization framework is designed for single-speaker interview videos and delivers significant positive societal impact by enabling efficient and contextually relevant video analysis that reduces the time needed to watch and review lengthy content. This reduction in viewing time supports well-being by lowering cognitive load and freeing users to focus on more meaningful tasks. In educational settings, the framework enhances accessibility and inclusiveness by providing concise summaries of lectures and interviews, thereby improving learning efficiency. In professional and media environments, it streamlines content curation and boosts productivity by saving hours of manual review, contributing to better work-life balance and employee satisfaction. By advancing human-centric AI with multimodal behavioural understanding, our work aligns with several United Nations Sustainable Development Goals (SDGs): SDG 4 (Quality Education) through enhanced accessibility to educational content; SDG 8 (Decent Work and Economic Growth) by improving workplace productivity and well-being; SDG 9 (Industry, Innovation and Infrastructure) through technological advancement in media analysis; and SDG 12 (Responsible Consumption and Production) by optimizing digital content consumption and reducing unnecessary resource expenditure.

We acknowledge potential challenges, such as biases from limited dataset diversity, privacy concerns related to sensitive audiovisual data, and risks of misuse through manipulated summaries. To mitigate these, we recommend fairness-aware training with diverse data, privacy-preserving techniques, and responsible development practices. While foundational in nature, this work highlights the importance of developing behaviour-aware video summarization technologies that maximize societal benefits, especially time savings and improved well-being, while carefully addressing associated risks.

## A.9 THE USE OF LARGE LANGUAGE MODELS (LLMs)

This research utilized LLMs, specifically GPT-4.5, GPT-3.5, and LLaMA-3.2 3B, to generate pseudo-ground truth summaries for evaluating our multimodal video summarization framework and to enhance the clarity of the manuscript. All LLM-generated outputs were rigorously evaluated and extensively revised by the authors to ensure alignment with academic standards and research objectives. LLMs did not contribute to research ideation, experimental design, or scientific analysis. The conceptual framework, experimental methodology, and conclusions are the original work of the authors. The authors assume full responsibility for all content, ensuring ethical use of LLMs and adherence to academic integrity standards.

