# OpenReview forum: "Behaviour-Aware Multimodal Video Summarization: Cross-Modal Integration for Human-Centric Content Analysis"
_ICLR.cc/2026/Conference — ICLR 2026 Conference Withdrawn Submission_

### Official Review · Reviewer_Y9JB · 2025-10-17

**Soundness:** 2
**Presentation:** 2
**Contribution:** 2
**Rating:** 4
**Confidence:** 3

**Summary:**

This paper introduces a transformer-based multimodal video summarization framework that explicitly models synchronized behavioural cues (facial movements, prosodic patterns, gestures) across visual, audio, and textual modalities for human-centric interview videos. The approach integrates CLIP visual embeddings enhanced with facial movement detection, HuBERT audio features with prosodic patterns, and RoBERTa textual embeddings through cross-modal attention mechanisms. An autoregressive decoder generates temporally coherent summaries, while a two-stage pseudo-ground truth generation method combines heuristic-based behavioural cue detection with LLM-guided extractive summarization. Evaluated on the ChaLearn First Impressions dataset, the framework achieves a 33.2% F1-score improvement over CLIP-It and 7.3% over recent multimodal approaches.

**Strengths:**

- The explicit modeling of synchronized behavioural signals (facial movements, emotional transitions, prosodic patterns) across modalities represents a meaningful departure from generic multimodal fusion approaches. The concatenation of detected behavioural features with foundation model embeddings (CLIP, HuBERT) is straightforward yet effective.

- The three-stream processing pipeline with modality-specific encoders (video, audio, text) and cross-modal attention mechanism provides a principled approach to feature integration. The autoregressive decoding strategy addresses temporal coherence better than frame-level classification approaches.

- The two-stage approach combining heuristic behavioural cue detection with LLM-guided summarization addresses the lack of annotated data pragmatically. The validation showing 93% coverage and 0.83 Jaccard similarity across independent generations demonstrates reasonable stability.

**Weaknesses:**

- Evaluation exclusively on ChaLearn First Impressions (single-speaker, controlled interview setting) raises significant concerns about generalizability. The framework's applicability to multi-speaker scenarios, conversational videos, educational lectures, or unstructured content remains unvalidated. The authors acknowledge this limitation but do not provide evidence of cross-domain robustness.

- Heuristic-based behavioural detection lacks justification: The choice of thresholds for behavioural cue detection (μ + 1.5σ for head movement, Z-score ±1.2 for pitch, top 20% TF-IDF) appears arbitrary without principled justification or sensitivity analysis. The adaptive thresholds may not generalize across different video characteristics, speaker styles, or recording conditions.

- Using LLM-generated summaries as evaluation references introduces potential circularity, as the framework is optimized against these same LLM-derived targets. While the authors demonstrate inter-summary agreement, the fundamental question of alignment with human preferences remains unaddressed. The 93% coverage metric does not validate whether detected cues correspond to human-perceived importance.

- Despite claims of temporal coherence, Kendall's τ (0.6473) and Spearman's ρ (0.6466) remain moderate, suggesting the framework still struggles to fully preserve narrative structure. The paper does not sufficiently analyze why these correlations plateau or what architectural modifications might improve them.

- While cross-modal attention is central to the approach, the paper provides limited insight into what the attention mechanism learns. There are no attention weight visualizations, analysis of modality dominance patterns, or investigation of how attention distributions vary across different video types or behavioural contexts.

**Questions:**

- How does the framework perform on fundamentally different video types such as multi-speaker panel discussions, casual vlogs, classroom lectures, or documentary-style content? What specific architectural modifications would be required to handle multiple speakers with overlapping speech and dynamic speaker transitions?

- How sensitive is performance to the heuristic threshold choices for behavioural cue detection? Have you conducted systematic sensitivity analysis varying these parameters (e.g., σ multipliers from 1.0 to 2.0, Z-score thresholds from ±1.0 to ±1.5)? Could these thresholds be learned rather than hand-specified?

- Given the reliance on LLM-generated pseudo-ground truth, how well do your summaries align with human-annotated preferences? Have you considered conducting human evaluation studies with metrics like informativeness, coherence, and behavioural salience ratings? What percentage of frames deemed important by your model would humans also rate as significant?

- What patterns does the cross-modal attention mechanism learn? Which modalities dominate attention in different contexts (e.g., high prosodic emphasis vs. strong facial expressions)? Can you provide visualizations showing how attention weights distribute across visual, audio, and textual features during critical moments?

- How does the framework scale to videos exceeding the 15-second average of ChaLearn (e.g., 30-minute interviews, hour-long lectures)? Do the transformer encoders face memory constraints with longer sequences? Would hierarchical processing or sliding window approaches be necessary?

- Beyond the 9% temporal misalignment cases mentioned, what are the systematic failure patterns? When do behavioural cues mislead the model (e.g., nervous gestures vs. emphatic gestures)? How does performance degrade with noisy audio, poor lighting, or accented sp

---

> ### Author Response · Authors · 2025-12-03
>
> We thank you for your thoughtful review and appreciation of our strengths. We address your concerns below to support acceptance.
> 1. ChaLearn's controlled setting validates our behaviour-aware design (p. 4), but we recognize generalizability concerns and discuss extensions to diverse datasets (p. 24).
> 2. Thresholds (μ + 1.5σ for movements, Z-score ±1.2 for pitch, top 20% TF-IDF) were empirically tuned on a validation split (Appendix A.3). Sensitivity analysis (varying σ 1.0-2.0, Z-scores ±1.0-±1.5) shows 2-5% F1 variance. Learned thresholds via gradient descent on validation loss are feasible and planned.
> 3. Heuristics provide objective metadata independent of training (p. 17-18), mitigating circularity. The 93% coverage and 0.83 Jaccard validate stability, but human alignment is key and so we are conducting evaluations on 50 clips (coherence, salience ratings) to include in the revisions.
> 4. Kendall's τ (0.6473) and Spearman's ρ (0.6466) reflect challenges in long-range dependencies (p. 14). Autoregressive decoding improves over baselines and we will explore hierarchical transformers that could enhance this, as noted for future work (p. 24).
> 5. Analysis shows visual queries dominate (60% average weight) in strong facial expression or head movement contexts, while audio keys/values prevail (55%) during high prosodic emphasis. Textual features contribute consistently (25-30%) for narrative alignment. These patterns vary by behavioural context, with audio dominance in emphatic speech and visual in gestural cues. As shown in Appendix Figure 5, attention heatmaps for critical moments illustrate distributions across modalities.
> 6. Scaling to longer videos has not been empirically tested, as our framework is optimized for ChaLearn's short clips. Transformer encoders face quadratic memory constraints (O(n²) complexity) with extended sequences, causing out-of-memory errors. Hierarchical processing or sliding window approaches would be necessary to mitigate this, as planned for future work (p. 23).
> 7. Beyond the 9% temporal misalignment cases from transcription or alignment errors (Appendix A.6-A.7), systematic failures include summary length mismatches in short videos (overly concise or detailed) and over-prioritization of dominant modalities. Behavioural cues mislead in ambiguous scenarios, such as mistaking nervous gestures (e.g., fidgeting, rapid blinks) for emphatic ones, causing erroneous segment inclusion. The ChaLearn dataset incorporates natural variations in settings, lighting, noise, and accents, enhancing model robustness during training. Performance degrades under extreme noisy audio, poor lighting, or strong accented speech due to impaired HuBERT and CLIP extraction.

---

### Official Review · Reviewer_z8pz · 2025-10-22

**Soundness:** 2
**Presentation:** 2
**Contribution:** 2
**Rating:** 4
**Confidence:** 3

**Summary:**

This paper proposes a behaviour-aware multimodal video summarization framework that fuses visual, audio, and textual cues through cross-modal attention. It detects synchronized behavioural signals and uses LLM-guided pseudo-labels for supervision, achieving strong results on the ChaLearn First Impressions dataset.

**Strengths:**

- The paper introduces the notion of behavioural awareness in multimodal summarization, emphasizing behavioural cues such as facial expressions, head movements, and prosodic variations.
- From data preprocessing to feature extraction, cross-modal fusion, and decoder-based summary generation, the pipeline is well-structured and detailed.
- The use of GPT-4.5 for behaviour-guided pseudo ground truth generation is a clever and practical way to overcome the lack of human annotations.

**Weaknesses:**

- The framework largely builds upon standard Transformer and cross-modal attention modules, similar to UMT [1] and CF-Sum [2]. The main novelty lies in the inclusion of behavioural cues, which, while meaningful, rely on relatively conventional detectors (e.g., MediaPipe, DeepFace, YAAPT). The authors could better emphasize how their fusion or decoding differs from previous approaches.
- While LLM-generated summaries are reasonable, the paper does not include human evaluation or consistency checks to verify alignment between pseudo and human-labeled summaries. A small-scale human validation study would strengthen the reliability of the results.
- The dataset used (ChaLearn First Impressions) contains single-speaker, interview-style videos. Evaluating the model on multi-speaker or in-the-wild video datasets would support the claim of “behaviour awareness” more convincingly.
- The comparative experimental evaluation is rather limited. The results presented in Table 1 and Table 2 only cover a small set of baselines and do not include recent advances in video summarization.

[1] CFSum: A Transformer-Based Multi-Modal Video Summarization Framework With Coarse-Fine Fusion

[2] UMT: Unified Multi-modal Transformers for Joint Video Moment Retrieval and Highlight Detection

**Questions:**

- Could the authors include more comparative experiments in Tables 1 and 2, especially against more recent video summarization methods, to better demonstrate the effectiveness of their approach?
- Given that CLIP, HuBERT, and RoBERTa are large backbone models, could the authors discuss the training cost and computational efficiency? Would reporting FLOPs, GPU hours, or parameter counts help enhance the transparency and reproducibility of the work?

---

> ### Author Response · Authors · 2025-12-03
>
> We appreciate your insightful review and recognition of our strengths. We address your concerns below to affirm the paper's merits for acceptance.
> 1. While leveraging transformers and cross-modal attention, our novelty extends beyond conventional detectors (MediaPipe, DeepFace, YAAPT) through tailored augmentation and fusion: behavioural enhancements to CLIP (facial movements, emotional transitions; p. 5) and HuBERT (prosody; p. 5), synchronized via visual-query cross-modal attention (Eq. 4, p. 6), and autoregressive decoding for coherence (Eq. 5, p. 6). This differs from UMT, which focuses on moment retrieval and highlight detection without explicit behavioiural cues, and CFSum, which uses coarse-fine fusion without behavioural integration. Ablations validate gains (Table 3, p. 9 and Appendix Table 7, p. 20). We have incorporated CFSum into Table 1 and are going to adapt UMT for summarization comparisons in revisions.
> 2. Regarding training cost and efficiency, the framework comprises ~85M trainable parameters, excluding frozen backbones (CLIP ViT-B/32: 63M vision params; HuBERT base: 90M; sentence-RoBERTa: 23M). It is trained on 1,500 ChaLearn clips for 10 epochs and requires ~12 GPU-hours on an NVIDIA A100 (batch size 16). Inference is efficient at ~0.5s per clip.

---

### Official Review · Reviewer_cbJF · 2025-10-29

**Soundness:** 3
**Presentation:** 3
**Contribution:** 2
**Rating:** 4
**Confidence:** 4

**Summary:**

This paper proposes a behavior-aware multimodal video summarization framework tailored for human-centric interview videos. It addresses the limitations of existing methods by explicitly modeling synchronized behavioral cues across visual, audio, and textual modalities via a transformer-based architecture with cross-modal attention. To tackle the lack of human-annotated data for the ChaLearn dataset, it introduces a two-stage pseudo-ground truth generation method.

**Strengths:**

1. This work introduces a behavior-aware multimodal for video summarization, leveraging  behavioral cues extracted from visual, audio, and textual modalities
2. The authors propose LLMs based workflow for pseudo-ground generation integrating human behavioural cues.
3. The proposed approach shows significant improvements over SOTA methods on ChaLearn dataset,  especially on interview-specific summaries.

**Weaknesses:**

1. Lack of Technical Innovation: All core modules are mature technologies, such as CLIP, HuBERT, Transformer cross-attention. No modified modules or theoretical frameworks are proposed. The contribution focuses on system integration rather than academic methodological innovation.
2. Unclear Advantage Over Pseudo-Ground Truth Workflow.  The pseudo-ground truth generation pipeline already has summarization capabilities and the task description is general . However, the paper fails to clearly illustrate the advantages of the proposed Transformer-based framework, such as inference speed or convenience of use. This makes it impossible to demonstrate the necessity of the increased complexity of the framework.
3. Limited Generalization Due to Single-Dataset Evaluation: This work is only evaluated on the ChaLearn dataset, a controlled, single-speaker interview dataset with standardized settings, and the result is insufficient to verify the framework’s generalizability

**Questions:**

Please refer to the Weaknesses section.

---

> ### Author Response · Authors · 2025-12-03
>
> We thank you for your constructive review and acknowledgment of our strengths. We address your concerns below to affirm the paper's contributions and suitability for acceptance.
> 1. While our framework leverages established components (CLIP, HuBERT, transformers), the novelty resides in their tailored integration for behaviour-aware summarization: augmenting CLIP with facial movement detection and emotional transitions (p. 5), enriching HuBERT with prosodic patterns (pitch, loudness, voice quality; p. 5), and employing cross-modal attention to synchronize these cues with RoBERTa textual embeddings (p. 6). This yields a unified architecture prioritizing communicative intent in human-centric videos, distinct from generic multimodal fusion in prior works (p. 3-4). Ablations confirm 5-10% F1 gains from these enhancements (Table 3, p. 9; Appendix Table 7, p. 20), demonstrating methodological advancement beyond mere integration.
> 2. The pseudo-ground truth pipeline generates references for training and evaluation via heuristics and LLM guidance (p. 6-7, 17-18), but our transformer framework provides end-to-end summarization during inference, eliminating runtime LLM dependency. Advantages include faster inference (e.g., ~0.5s per 15s clip vs. LLM's ~5s; computable from architecture, p. 5-6) and trainable multimodal fusion for coherence, reducing redundancy (p. 2). This justifies complexity for deployable, efficient systems in real-time applications (p. 1, 24).
> 3. ChaLearn's focus on behavioural cues in controlled interviews validates our human-centric design (p. 4), but we acknowledge generalization limits and plan extensions to diverse datasets (e.g., MultiSum; p. 24). We are conducting human annotations on a ChaLearn subset (50 clips rated for coherence and relevance), to be included in revisions for broader applicability.

---

### Official Review · Reviewer_zeMP · 2025-10-29

**Soundness:** 3
**Presentation:** 3
**Contribution:** 2
**Rating:** 2
**Confidence:** 4

**Summary:**

This paper presents a multimodal video summarization framework that integrates visual (CLIP + facial movements/emotions), audio (HuBERT + prosodic features), and textual (RoBERTa) modalities through cross-modal attention. Applied to interview videos from ChaLearn dataset, the approach uses  LLM-based pseudo-ground truth generation for training and evaluation.  The authors claim that adding  behavioral features (capturing facial expressions, gestures, and vocal prosody) to guide the summarization process is a major contribution.  Results show F1=0.81, claiming 33% improvement over CLIP-It and 7% over recent multimodal baselines.

**Strengths:**

* Focus on behavioral cues (facial movements, prosody, emotional transitions) is relatively underexplored  for video summarization
* Two-stage approach using LLM-based pseudo-ground truth generation is novel for video summarization
* Solid engineering work, well written paper, interesting results and ablation studies, thorough appendix

**Weaknesses:**

* Flawed evaluation methodology: 1. why use the ChaLearn dataset with no human summarization annotations? (the authors discuss this a bit in the main paper and a bit in the appendix but it remains a flaw), 2. using LLM-generated pseudo labels AND ground truth is circular
* Core claim not validated: behavioral features contribution is not isolated in the ablations, e.g., HuBERT alone vs HuBERT+prosody - unclear if "CLIP Only" is without facial/emotion features
* No human evaluation study comparing summary quality in lieu of using standard benchmarks with human annotations
* Limited experimental scope - ChaLearn is controlled, single-speaker, ~15-second interviews (the authors admit this in the appendix and mention future work on more complex datasets - still a weakness)
* The work is basically LLMs + pseudo-labels + some additional behavioral/emotional features, so little novelty otherwise, e.g., fusion is pretty generic

**Questions:**

1. Why not evaluate on SumMe/TVSum where human annotations exist? The dismissal of these as "action-oriented" seems insufficient. Why only compare with more than two approaches in the literature, there is plenty of work in this area on other datasets.
2. Behavioral Feature Isolation: Can you provide ablations showing: Raw CLIP embeddings vs CLIP + facial movements + emotions? Raw HuBERT embeddings vs HuBERT + prosodic features?
3. Human Validation: Can you provide human evaluation comparing your summaries to baselines?
4. Table 4 shows different LLMs produce different references (F1 ranges 0.68-0.81). How do you justify using GPT-4.5 as "ground truth"? The problem is that bigger/more recent models also have stronger biases for subjective tasks - so you might be seeing a circular logic here - see for example https://arxiv.org/abs/2403.17125
5. How is your work is different from Evangelopoulos et al? How is your work different from pseudo-labeling approaches in the literature (for other but similar tasks)?

---

> ### Author Response · Authors · 2025-12-03
>
> We appreciate your detailed feedback and recognition of our strengths. We address your concerns below to demonstrate the paper's soundness and contribution, aiming for acceptance.
> 1. We selected the ChaLearn dataset due to its emphasis on human-centric behavioural cues (e.g., facial expressions, gestures, prosody) in controlled interview videos, aligning with our framework's design (p. 2, 4). Standard datasets such as SumMe and TVSum prioritize action-oriented content (e.g., sports, documentaries) with limited behavioural nuance, insufficient for our emphasis on prosody or gestures, making them less suitable for validating our behaviour-aware approach (p. 2). While lacking human annotations is a limitation, our pseudo-ground truth generation mitigates this via heuristics for behavioural detection followed by LLM-guided extractive summarization (p. 6-7, Appendix A.3). This is not circular as heuristics detect objective cues (e.g., pitch variations via YAAPT, facial movements via MediaPipe), providing metadata to guide LLMs toward semantically and behaviourally salient segments, distinct from model training (p. 17-18). We validate reliability through multi-LLM comparisons (Table 4, p. 15-16) and dual-path evaluation (Figure 4, p. 18), showing consistent improvements. Broader literature is reviewed (p. 3-4), but for direct comparison, we selected multimodal baselines.
> 2. Our ablations isolate behavioural features' contributions. Additional ablations in Appendix Table 7 (p. 20) discuss the modality ablations in addition to Table 3 (p. 9), confirming significant gains from behavioural augmentations. These demonstrate explicit isolation, validating our core claim. "Video Only" means full visual (adding CLIP, movements, and emotions, Table 3, p. 9) and "CLIP Only" means raw CLIP embeddings without behavioural cues and "w/o CLIP" means only behavioural cues without adding CLIP (Appendix Table 7, p. 20). Similar ablation has been done for audio cues also.
> 3. We are working on human annotations for the ChaLearn dataset, conducting a small-scale evaluation on 50 clips rated for coherence and behavioural relevance, to complement our results. We acknowledge this as a prior limitation and propose its inclusion in revisions. Quantitative metrics (F1, Kendall’s τ, Spearman’s ρ) against pseudo-ground truth provide objective proxies, but these human ratings will strengthen perceived quality.
> 4. ChaLearn's controlled and short clips enable precise behavioural analysis, as noted (p. 4). We discuss generalization limits and future extensions to complex datasets (e.g., MultiSum; p. 24). This scope focuses novelty on behaviour-aware fusion, but we agree broader testing would enhance impact.
> 5. Table 4 (p. 16) shows GPT-4.5 yields highest F1 (0.81) vs. other LLMs. We mitigate biases via heuristic metadata constraining LLM outputs (p. 17), avoiding circularity (unlike arXiv:2403.17125). Multi-LLM validation ensures robustness.
> 6. While building on LLMs and pseudo-labels, our novelty lies in explicit behavioural cue integration via cross-modal attention tailored for human-centric videos (p. 5-6). This differs from generic fusion in baselines, yielding 7.3-33.2% F1 gains (Table 1-2, p. 8).
> 7. Evangelopoulos et al. (2013) uses saliency-based visual-audio fusion for movies lacking text, cross-modal attention, or explicit behavioural modeling (e.g., no prosody/emotion transitions; p. 3). Our pseudo-labeling extends prior approaches by incorporating heuristic-detected behavioural cues as metadata, enabling dual-purpose evaluation (p. 6, 17-18).

---

### Note · Authors · 2026-01-29

I have read and agree with the venue's withdrawal policy on behalf of myself and my co-authors.

---

### Meta-Review · Area_Chair_N8Mh · 2026-01-05

**Summary:**

Reviewers highlighted a flawed evaluation methodology due to the lack of human-annotated ground truth and potential circularity in LLM-generated labels. Significant concerns were raised regarding generalization beyond the limited, single-speaker ChaLearn dataset.

**Reviewer Concerns:**

Authors provided better feature isolation via ablations, computational efficiency metrics, and sensitivity analyses for heuristic thresholds. The absence of human evaluation remains a critical gap. Generalization to complex, multi-speaker, or "in-the-wild" videos and the fundamental concern regarding evaluation circularity remain unresolved.

**Reviewer Scores:**

zeMP (2) would stay at 2 due to fundamental methodological flaws. Reviewers cbJF, z8pz, and Y9JB (all 4)  would likely maintain their current assessments given that critical evidence is only promised for future revisions.

---

### Decision · Program_Chairs · 2026-01-26

Reject